# New Era of Immunotherapy in Pediatric Brain Tumors: Chimeric Antigen Receptor T-Cell Therapy

**DOI:** 10.3390/ijms22052404

**Published:** 2021-02-27

**Authors:** Wan-Tai Wu, Wen-Ying Lin, Yi-Wei Chen, Chun-Fu Lin, Hsin-Hui Wang, Szu-Hsien Wu, Yi-Yen Lee

**Affiliations:** 1School of Medicine, National Yang Ming Chiao Tung University, Taipei 112304, Taiwan; peter82102057@gmail.com (W.-T.W.); chenyw@vghtpe.gov.tw (Y.-W.C.); cf_lin@vghtpe.gov.tw (C.-F.L.); 2Division of Pediatric Neurosurgery, Department of Neurosurgery, Neurological Institute, Taipei Veterans General Hospital, 201, Section 2, Shih-Pai Road, Taipei 112201, Taiwan; 3Department of Internal Medicine, Taipei Veterans General Hospital, Taipei 112201, Taiwan; sincesnow@gmail.com; 4Division of Radiation Oncology, Department of Oncology, Taipei Veterans General Hospital, Taipei 112201, Taiwan; 5Department of Neurosurgery, Neurological Institute, Taipei Veterans General Hospital, Taipei 112201, Taiwan; 6Department of Pediatrics, Division of Pediatric Immunology and Nephrology, Taipei Veterans General Hospital, Taipei 112201, Taiwan; hhwang@vghtpe.gov.tw; 7Department of Pediatrics, Faculty of Medicine, School of Medicine, National Yang Ming Chiao Tung University, Taipei 112304, Taiwan; 8Institute of Emergency and Critical Care Medicine, School of Medicine, National Yang Ming Chiao Tung University, Taipei 112304, Taiwan; 9Department of Plastic and Reconstructive Surgery, Taipei Veterans General Hospital, Taipei 112201, Taiwan; shwu3@vghtpe.gov.tw; 10Department of Surgery, School of Medicine, National Yang Ming Chiao Tung University, Taipei 112304, Taiwan

**Keywords:** chimeric antigen receptor (CAR) T cells, gene modified-based cellular platform, immunotherapy, pediatric brain tumor

## Abstract

Immunotherapy, including chimeric antigen receptor (CAR) T-cell therapy, immune checkpoint inhibitors, cancer vaccines, and dendritic cell therapy, has been incorporated as a fifth modality of modern cancer care, along with surgery, radiation, chemotherapy, and target therapy. Among them, CAR T-cell therapy emerges as one of the most promising treatments. In 2017, the first two CAR T-cell drugs, tisagenlecleucel and axicabtagene ciloleucel for B-cell acute lymphoblastic leukemia (ALL) and diffuse large B-cell lymphoma (DLBCL), respectively, were approved by the Food and Drug Administration (FDA). In addition to the successful applications to hematological malignancies, CAR T-cell therapy has been investigated to potentially treat solid tumors, including pediatric brain tumor, which serves as the leading cause of cancer-associated death for children and adolescents. However, the employment of CAR T-cell therapy in pediatric brain tumors still faces multiple challenges, such as CAR T-cell transportation and expansion through the blood–brain barrier, and identification of the specific target antigen on the tumor surface and immunosuppressive tumor microenvironment. Nevertheless, encouraging outcomes in both clinical and preclinical trials are coming to light. In this article, we outline the current propitious progress and discuss the obstacles needed to be overcome in order to unveil a new era of treatment in pediatric brain tumors.

## 1. Introduction of Pediatric Brain Tumors

Primary malignant central nervous system (CNS) tumors, including medulloblastomas, ependymomas, astrocytomas, and germ cell tumors, serve as the second most common pediatric malignancies, just after hematological cancers [1]. Nevertheless, they last as a main reason of pediatric cancer-related death [2]. Among them, more than 90% are located in the brain, with an incidence of 1.12–5.14 cases per 100,000 children [3]. While the etiology of childhood brain tumors remains unclear, it has been proposed that genetic factors, environmental factors, family history, parental age at birth and cancer predisposition syndromes might be related [4].

Currently, surgical resection, chemotherapy, and radiotherapy are the major therapeutic strategies for pediatric brain tumors [5]. Even though chemotherapy and radiotherapy are more effective in pediatric patients with brain tumors than their adult counterparts [6], significant neurologic deficits and neurocognitive morbidities which impede future ability to live independently are concern [7]. Under these circumstances, immunotherapy, which only selectively destroys malignant cells expressing target antigen while leaving healthy tissues undamaged, may be a valuable therapeutic option. Chimeric antigen receptor (CAR) T-cell therapy, in particular, targeting tumor-specific antigens via genetically modified T cells, might be more useful for pediatric brain tumors, as they are well-known for the lack of high somatic tumor mutational burden [8,9].

## 2. Chimeric Antigen Receptor (CAR) T-Cell Therapy

CARs are artificially synthesized proteins which incorporate three major components: an extracellular tumor-specific antibody, an intracellular signaling motif, and a transmembrane domain serving as a bridge [10,11]. The outermost part is responsible for antigen targeting, as it possesses a single-chain fragment (scFv) purified from antibody that is specific for tumor antigens [12]. This part of CARs is responsible for being bound to tumor cells and triggering consequent T-cell activation and proliferation, and inaugurates cytokine release and cytolytic degranulation [13]. As for intracellular domain, it determines the strength, quality, and persistence of a T-cell response to tumor antigens [14], and it is frequently manipulated to enhance the potency of CAR T-cell therapy. To date, there are five generations of CARs being developed. The endodomain of the first generation of CARs comprises the CD3-ζ chain alone, with limited T-cell expansion and insufficient cytokine release [15]. Under this consideration, the second generation of CARs incorporated costimulatory domain, either CD28 [16,17] or 4-1BB [18], intracellularly, to ameliorate T-cell proliferation and persistence [19]. The third generation combined CD28 and 4-1BB [20,21], to further increase T-cell expression and persistence. Meanwhile there are two main immune systems, namely innate immune and adaptive immune in human body. Innate immune system, which serves as the first line of immune response and is antigen-independent, is thought to be helpful in adaptive T-cell therapy. Thus, another modulation combining innate immune response with CAR T cells was proposed. The recent fourth generation added cytokines, such as interleukin-12 (IL-12), to the endodomain of the second generation; they which could activate T cells, as well as natural killer cells, simultaneously, when encountering tumor cells [22]. A natural killer cell is capable of appealing cytokine cassette and inducing cytotoxicity against tumor cells [23]. This combination allows antigen-negative cancer cells to be eliminated concurrently. This mergence was termed T cell redirected for universal cytokine-mediated killing (TRUCKs) [19]. In this situation, tumor microenvironments are amended, and the lifespan of CAR T cells is prolonged. The clinical trials of this concept combining innate and adaptive immunity are still in cradle [24]. By synchronous installation of IL-2 receptor and binding site for the transcription factor STAT3 to the endodomain, vigorous JAK–STAT3/5 cytokine, cascade can be instigated in local tumor environment and thus minimizes systemic inflammation [25] (Figure 1). The contemporary protocol of CAR T-cell therapy adopted autologous T cells (Figure 2).

## 3. Application of CAR T-Cell Therapy, from Hematological Malignancies to Pediatric Brain Tumors

In 2012, the first child received CD19-targeted CAR-T therapy for her relapsed B-cell acute lymphoblastic leukemia exhibited complete remission and no refractory or relapse for more than five years [26]. This finding opened up a new era of CAR-T therapy for malignancies. Afterwards, several studies demonstrated promising response, ranging from 60% to 93% complete remission rate, with minimal residual disease (MRD)-negative of CAR-T therapy for pediatric hematological malignancies [27,28,29,30]. The first CAR-T therapy, tisagenlecleucel, was approved by the FDA in August 2017 for refractory or relapsed acute lymphoblastic leukemia in patients younger than 25 years old [31,32,33]. In October of the same year, axicabtagene ciloleucel was approved for refractory or relapsed diffuse large B-cell lymphoma as well [34]. These striking successes may be due to specific homogeneous tumor target antigens in B-cell lineages [35]. With the encouraging results in hematological malignancies, CAR-T therapy was used to treat a variety of solid tumors. However, the response to solid tumors was not as effective as that of hematological malignancies [36]. The possible reasons include heterogeneous and low specific target antigen expression on tumor surface, insufficient CAR T cells traveling to and infiltrating into the tumor, limited T-cell expansion, and poor persistence because of the immunosuppressive tumor microenvironment [35,37]. Brain tumors are notorious for their immunosuppression environment, possibly due to the unique composition of the extracellular matrix; distinctive tissue-resident cell types, such as astrocytes, which are known to blunt cytotoxicity; and a natural inflammation shelter/blood–brain barrier (BBB) [38]. Furthermore, possible on-target off-tumor toxicity of CAR-T therapy may reduce the cytotoxic effect on tumor cells and may increase potential treatment-related toxicities on normal tissues [39]. Nevertheless, several clinical and preclinical studies have shown favorable efficacy in solid tumors, especially anti-carcinoembryonic antigen (CEA) therapy, including CD3ζ, CD28–CD3ζ, and locally administered CAR T cells [40].

Zhang et al. demonstrated that 7 of 10 patients with metastatic colorectal cancer refractory to standard treatments became stable disease (SD) from progressive disease (PD) after undergoing treatment with CAR T cells [41]. Thistlethwaite and her colleagues also reported that 7 of 14 relapsed and refractory metastatic gastrointestinal patients achieved stable disease and persisted for six weeks after CAR T-cells infusion. One of the patients even stayed alive for 56 months [42]. In other kinds of malignancies, such as high-risk osteosarcoma, Chulanetra et al. proved that CAR T cells have synergistic effect with doxorubicin on eliminating tumor cells of osteosarcoma [43]. Some patients developed transient side effects such as acute respiratory toxicity, but no severe irreversible toxicity was observed in patients underwent treatment [44]. The outcomes of these abovementioned trials support the efficacy and safety of the CAR-T therapy. Therefore, more and more clinical trials aim to achieve the promising results of application in pediatric solid tumors, especially brain tumors [45,46,47,48]. Though medical technology has improved largely in the past few decades, treatments for brain tumors are still disappointing [49,50]. Highly specific and personalized treatments such as CAR T-cell therapy offer an opportunity to fight against pediatric brain tumors [51].

CAR T cells against tumor-specific antigens, including epidermal growth factor receptor variant III (EGFRvIII), human epidermal growth factor receptor 2 (HER2), and interleukin 13 receptor alpha 2 subunit (IL13Rα2) of glioblastoma (GBM), were under clinical studies to collect data about their safety and feasibility in recent years. The routes of administration, such as intravenous and intratumor infusion, were also evaluated in these researches. Meanwhile, many preclinical trials are also being conducted to find out more possibilities of CAR T-cell therapy in pediatric brain tumors. Further details are discussed below.

## 4. Preclinical Results of CAR T-Cell Therapy in Pediatric Brain Tumors

### 4.1. Current Preclinical Results

The targeted antigens of CAR T cells should be tremendously expressed on cancer cells but not on normal tissues to have the highest efficacy [52]. An orthotopic xenogeneic mouse model of medulloblastomas revealed tumor regression, using even the first generation of CAR T-cell therapy targeting HER2 [53]. The application of the second-generation CAR T cells further demonstrated improved response and durable regression and promised practicability and safety in non-human primates via intraventricular delivery [54]. In pediatric medulloblastoma, it has been revealed that B7-H3 CAR T cells, by producing IFNg, TNFa, and IL-2, can have antitumor effects in xenograft models [55]. In both mice medulloblastoma and diffuse intrinsic pontine glioma (DIPG), the survival was extended remarkably by B7-H3 CAR T cells and exhibited minimal binding to healthy tissues [56]. Intracerebroventricular or intratumoral administration of B7-H3 CAR T cells also mediated antitumor effects against cerebral atypical teratoid/rhabdoid tumors xenografts in mice [57]. Approximately, preferentially expressed antigen in melanoma (PRAME), a tumor-associated antigen is expressed in around 80% of medulloblastomas [58], and it also exhibited promising results when engineered into CAR T-cell targets in orthotopic medulloblastoma models [59]. Recently, new targets of CAR T cells in glioma: podoplanin (PDPN), a member of type I transmembrane glycoproteins, as well as CD70, one of the tumor necrosis factor receptors have been recognized [60,61]. For H3-K27M-mutant diffuse midline gliomas (DMGs), which are mostly unresectable because of infiltration into the surrounding areas, Mount et al. [62] also demonstrated exceeding antitumor cytotoxicity of GD2-directed CAR T cells both in vitro and in vivo. Numerous preclinical trials and clinical trials of CAR T cell targeting GD2 for various CNS tumors are ongoing [21,63,64,65]. Currently, many clinical trials have exhibited that CAR T-cell therapy targeting GD2 is well tolerated in neuroblastoma, and we discuss this further below [21,63,65].

### 4.2. Identifications of Further Antigens

Patient-derived xenografts (PDXs) is a useful material to identify tumor antigens to serve as targets of CAR T-cell therapies. Using this method, CXCL5/CXCL6 genes were recognized to be overexpressed in malignant rhabdoid tumors [66]. PTK7, an intersection gene of WNT, VEGF, and stem cell function [67], could also be a potential target. Increased expression of PTK7 RNA was detected in atypical teratoid/rhabdoid tumors and repressed by PTK7 knockdown, as well as vatalanib, a tyrosine kinase inhibitor that blockades angiogenesis [68]. A recent study further established an expression hierarchy, B7-H3 = GD2 >> IL13Rα2 > HER2 = EphA2, using orthotopic xenografts derived from 49 patients [69], which might be of use for future design of immunotherapies.

### 4.3. Prevention of Tumor Antigens Escape

As heterogeneity of antigen expression is observed in most brain tumors, targeting multiple antigens is demonstrated to increase antitumor potency and lower the possibility of tumor antigen escape [70]. The use of trivalent targets to EphA2, HER2, and IL-13Rα2 of CAR T cells exhibited benefits in preclinical models of recurrent medulloblastoma, GBM, and ependymomas [71,72]. In a murine GBM model, simultaneously targeting HER2 and IL-13Rα2 showed better tumor control [70,73]. As a trivalent vaccine targeting EphA2, HER2, and IL-13Rα2 in pediatric malignant gliomas [74] and ependymomas [75] revealed feasibility, tolerability, and efficacy pediatric brain tumors, combination of targeted antigens might be the future direction of the development of CAR T-cell therapy.

## 5. Clinical Trials of CAR T-Cell Therapy in Pediatric Brain Tumors

### 5.1. Interleukin 13 Receptor Alpha 2 Subunit (IL13Rα2)

IL-13Rα2 is a plasma membrane receptor highly expressed on 50% to 78% GBM and associated with poor survival rate. On the other hand, IL-13, an anti-inflammatory cytokines secreted by CD4^+^ T cells, nature killer cells, mast cells, and eosinophils can signal through IL-13Rα2 to induce the progression of GBM [76,77]. The first in-human local administration of CAR T cells into the resected cavity of brain tumor was done in 2015 (NCT00730613). The repetitive intracranial infusions of IL-13Rα2 CAR T cells against recurrent high-grade glioma resulted in significantly increased necrotic lesions. The whole procedure was well tolerated, with limited and transient adverse events, such as brain inflammation [46]. This clinical trial confirmed that IL13Rα2 was a useful and safe immunotherapeutic target in GBM.

### 5.2. Epidermal Growth Factor Receptor Variant III (EGFRvIII)

Epidermal growth factor receptor (EGFR) is highly expressed in various tumors of which EGFRvIII is the most common variant due to an in-frame deletion of *EGFR* exons 2–7. Though EGFRvIII is therefore considered to be an oncogene, its characteristics including expression in about 30% of GBM cases, but absence in normal tissues make it a suitable target in CAR-T immunotherapy [78]. In 2017, the first-in-human infusion of EGFRvIII CAR-T therapy via intravenous route for recurrent glioblastoma in adolescents and adults was reported (NCT02209376). The results revealed median overall survival for about eight months, with the longest stable disease for at least 18 months. No cytokine release syndrome (CRS) or neurotoxicity was noted. These findings suggest that perhaps intravenous administration can pass through the blood–brain barrier and achieve adequate antitumor activity in the brain, with limited off-target toxicity [48]. However, another EGFRvIII trial, which was applied to 18 teenagers and adults by Goff and her colleagues, demonstrated median progression-free survival for only 1.3 months. Some participants developed adverse effects like dyspnea or hypoxia, which also led to treatment-related mortality in one patient, suggesting that a safer and more feasible protocol is needed [47].

### 5.3. Human Epidermal Growth Factor Receptor 2 (HER2)

HER2, a cell surface receptor expressed on numerous malignancies, such as breast cancer, ovarian cancer, and glioblastoma, is one of the members of EGFR family. The overexpression of HER2 is associated with poor survival. A previous case report had raised safety concerns about HER2 CAR-T therapy because of respiratory distress and even death after treatment. This might be due to HER2 is expressed in both tumor cells and normal tissues [20,79]. Nevertheless, recent clinical trial with CAR T-cells targeting HER2 demonstrated that 17 participants including pediatric patients with progressive HER2-positive glioblastoma achieved 11.1 months and 24.5 months of overall survival after CAR T-cells infusion and after diagnosis, respectively. One patient had partial responded, and seven patients achieved stable disease. No dose-limiting toxicity was found after infusion [45].

### 5.4. Other Ongoing Clinical Trials

An ongoing clinical trial with intraventricular administration of IL-13Rα2 CAR T cells for recurrent or refractory malignant glioma revealed significant tumor volume reduction in one patient, followed by complete response for more than 7.5 months [80]. Other targets for pediatric brain tumors include B7-H3, EGFR806, and GD2. B7-H3 is a transmembrane protein, as well as an immune checkpoint molecule expressed in several kinds of pediatric cancer, including diffuse intrinsic pontine glioma and medulloblastoma [81]. EGFR806-CAR T cells are CARs with mAb806-based binders, which recognize untethered EGFR on tumor tissues, including malignant glioma [82]. GD2 is one of the subtypes of the disialoganglioside. It is highly expressed in various type of pediatric tumors, such neuroblastomas, retinoblastomas, Ewing sarcomas, and gliomas, but shows minimal expression in normal tissues [83]. All of the above are regarded as the well-suited targets for cancer therapy and under recruitment in the ongoing pediatric clinical trials (Table 1).

## 6. Current Challenges of CAR-T Therapy in Pediatric Brain Tumors

### 6.1. Adverse Effects of the Central Nervous System

CAR T-cell therapy is eminent for its two major toxicities: cytokine releasing syndromes (CRS) and neurotoxicity. CRS is an acute systemic inflammatory response caused by hypersecretion of cytokines during immune reaction. The symptoms of CRS could be as easy as isolated fever, or potentially life-threatening, such as refractory hypotension or consumptive coagulopathy [84,85]. The severity of symptoms usually correlates with tumor burdens [86], and can be graded from one to four, based on the presence of fever, hypotension, hypoxia, end organ dysfunction, and admission to intensive care units [87,88]. Those symptoms can happen on the first day of the infusion of CAR T cells or can be delayed up to 14 days after the initiation of the delivery of CAR T cells. Anti-IL-6 receptor antibody, tocilizumab, has been demonstrated to reverse CRS [89]. Prophylactic administration of tocilizumab during CAR T-cell therapy is also undergoing clinical trial (NCT02906371). For neurotoxicity, namely immune effector cell-associated neurotoxicity syndrome (ICANS) or CAR-T-cell-related encephalopathy syndrome (CRES), various symptoms, including headache, confusion, neurological deficits, and even rarely cerebral edema, were reported [90]. In preclinical trials, the peritumoral edema caused by T-cell infiltration and consequent hydrocephalus even led to death in murine model [62]. These results highlight the significance of closely monitoring the neurological complications of CAR T-cell therapy henceforth. A meta-analysis of CAR T-cell trials for cancer has demonstrated around 55% and 37% of all patients experienced CRS and neurotoxicity, individually [8]. Although there is no specific analysis for brain tumor currently, intraventricular or loco-regional infusion of CAR T cells may mitigate these toxicities due to the lack of systemic response [52]. In one preclinical study, inflammation following CAR T-cell infusion led to brain swelling and hydrocephalus, which required emergent neurosurgery [91]. These results pictured CAR T-cell therapy as a double-edged sword. Further studies and close monitoring of CAR T-cell therapy in brain tumor are needed.

### 6.2. Limitations of CAR T-Cell Therapy in CNS Tumors

The applications of CAR T-cell therapy to pediatric brain tumors need to take not only the poor immune response of solid tumors but also the difficulties in delivery of T cells across the blood–brain barrier (BBB) into consideration. For solid tumors, on-target, off-tumor toxicity, tumor immunosuppressive microenvironment (TME), and antigen escape remain the major concerns (Figure 3). In contrast to lymphoid cancer, whose antigens are confined to B-cell lineage [92], solid tumors rarely have specific antigens on the cell surface. Even if the tumor antigens were identified, vital tissue might express the same antigens. Encouragingly, absence of toxicity was observed in limited expansion of CAR T cells in solid tumors [93]. Moreover, attuning the affinity of the scFv domain of CAR T cell might enable T cells to distinguish tumor cells from normal tissues [94]. Besides this, immune suppression from the TME in solid tumors [7,95], including GBM [96], was noted in several studies. In the human body, there are a bunch of immune checkpoints that serve as brakes to prevent T cells from attacking our own cells. However, cancer cells can also titillate those checkpoints to stave off immune activation. Under these circumstances, the efficacy of CAR T cells in the tumor microenvironment might be reduced. Thus, combinational use of immune checkpoint receptor inhibitors, such as PD-1/CTLA-4 inhibitors, might be beneficial [97,98]. Targeting extracellular components, such as VEGF or TGF-β, might also improve T-cell migration and expansion [99]. As for antigen escape, targeting multiple antigens simultaneously is currently under scrutiny for its efficacy and persistence [100]. To cross the BBB, local/regional infusion of CAR T cells directly into the tumor cavity or ventricular systems might be practical considering the promising results observed, but its safety requires further investigation [101,102]. Nanoparticles can also serve as a new platform for CAR T-cell delivery into the CNS [103]. Recently, an affibody molecule (ZSYM73) was discovered to potentially increase the penetration into the CNS via attaching to the transferrin receptor (TfR) [104], which might be applied to the transportation of CAR T cells into the CNS.

Finally, the relapsed brain tumors, as observed in leukemia as well [30,105], could be associated with antigen disguise, antigen density, or genetic mechanism [95], which call for future in vitro and in vivo studies to understand their biological characters fully. Challenges remain in combinations with other therapies, such as Lenalidomide used in hematological malignancies (NCT03070327) or cereblon-modulating agent CC-122 (NCT03310619) [106] to improve CAR T-cell potency. Memory T cells, with the capability of self-renew and the elasticity to differentiate into effective T cells once reencountering antigens, might be of use in CAR T-cell therapy, to treat minimal residual disease [107]. As the maturation mechanism of memory precursors T cells is gradually unveiled [108], which involves IL-7 and IL-15 [107] and is known to function through activation of STAT [109], further modification of the CAR structure, such as substituting IL-2 with IL-7 and IL-15 [107,110] or adding JAK-STAT to intracellular domains of CAR structure, might enrich the T memory stem cell population and prolong its persistence.

## 7. Conclusions

Pediatric brain tumors, distinct which are from their adult counterparts in both genetic mutations and expression, as well as treatment-related adverse effects on childhood growth in functions and capability, need novel therapies urgently. In the past decades, the understanding of CAR T-cell therapy has become more comprehensive, and its safety, efficacy, and persistence have gradually improved. With the advances of “multi-omic” profiling and CRISPR/Cas 9 genetic modulation, the novel tumor antigens are being developed, and the potency of CAR T-cell therapy is ameliorated. Future research investigating more suitable delivery of CAR T cells, minimizing off-tumor toxicity and dwindling the cost of CAR T-cell therapy, will shed light on the prognosis, as well as improve the quality of life in children with brain malignancies.

## Figures and Tables

**Figure 1 ijms-22-02404-f001:**
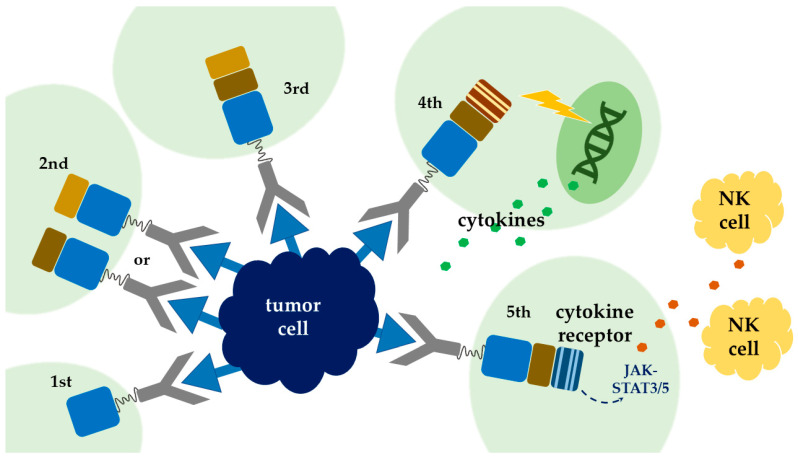
Five generations of chimeric antigen receptor (CAR) T-cell therapy. The blue cloudy shape in the middle symbolizes tumor cells with tumor antigens. The tumor is circled by five generations of CAR T cells. The five generations of CAR T-cell therapy are sequentially placed in clockwise direction. Gray Y type represents antigen-binding domains. The green shadow symbolizes intracellular space. The transmembrane domain serves as a bridge between the ectodomain and the endodomain. The first generation only contains CD3ζ as an intracellular domain, which is the sky-blue box in the figure. CD28 or 4-1BB is then added, to generate the second generation. The third generation consists of both CD28 and 4-1BB motifs. Genes encoding cytokines including IL-12 or IL18 are tethered to the intracellular domain, to develop the fourth generation. The fifth generation comprises the IL-2 receptor and binding site for the transcription factor STAT3, in order to induce cytokine storm.

**Figure 2 ijms-22-02404-f002:**
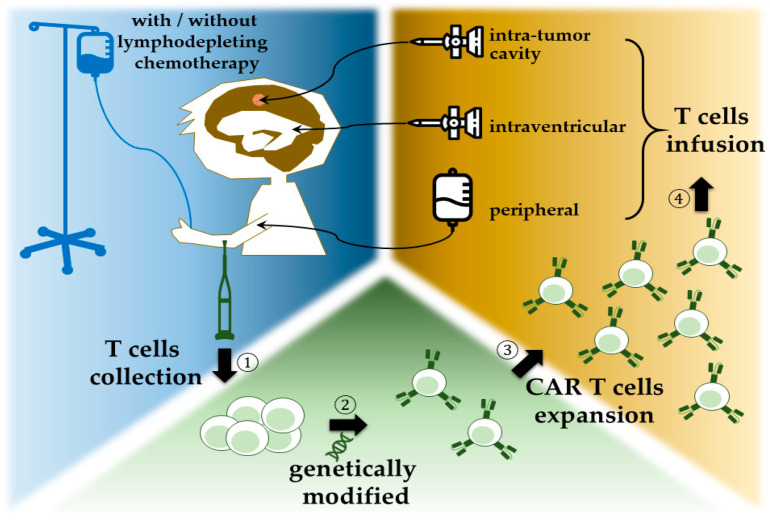
Protocol of CAR T-cell therapy: (1) T cells are gathered from patient. (2) CAR structure is engineered into the collected T cells. (3) The modified CAR T cells are augmented to sufficient amounts. (4) CAR T cells are infused back into the patient via either catheter into intratumor cavity or intraventricular, or through peripheral infusion.

**Figure 3 ijms-22-02404-f003:**
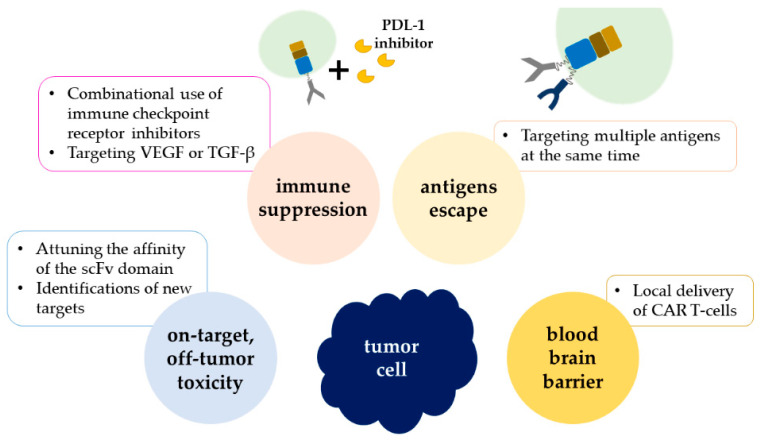
Descriptions of current limitations of CAR T-cell therapy in pediatric brain tumors. The figure lists the four current main obstacles in applying CAR T-cell therapy in pediatric brain tumors and the corresponding solutions.

**Table 1 ijms-22-02404-t001:** Published and ongoing clinical trials with CAR T-cell therapy for pediatric brain tumors.

Trial	Target	Phase	N	Age	Tumor Type	Outcome or Recruitment Status
NCT01109095 (Baylor College of Medicine)	HER2	I	17	10–69 years	Progressive HER2-positive glioblastoma	Median OS 11.1 months 1/16 PR, 7/16 SD (3/16 SD for 24–29 months)
NCT04185038 (Seattle Children’s Hospital)	B7-H3	I	70	1–26 years	Diffuse intrinsic pontine glioma/diffuse midline glioma and recurrent or refractory pediatric CNS tumors	Recruiting
NCT03638167 (Seattle Children’s Hospital)	EGFR806	I	36	1–26 years	EGFR-positive recurrent or refractory pediatric CNS tumors	Recruiting
NCT 04099797 (Baylor College of Medicine)	GD2	I	34	12 months–18 years	GD2-expressing brain tumors	Recruiting
NCT02442297 (Baylor College of Medicine)	HER2	I	28	3 years and older	HER2-positive CNS tumors	Recruiting
NCT03500991 (Seattle Children’s Hospital)	HER2	I	48	1–26 years	HER2-positive recurrent/refractory pediatric CNS tumors	Recruiting
NCT02208362 (City of Hope Medical Center)	IL-13Rα2	I	92	12–75 years	Recurrent or refractory malignant glioma	Recruiting (1/1 CR for 7.5 months)

CAR T cell = chimeric antigen receptor T cell; HER2 = human epidermal growth factor receptor 2; OS = overall survival; PR = partial response; SD = stable disease; B7-H3 = CD276; CNS = central nervous system; EGFR = epidermal growth factor receptor; GD2 = disialoganglioside; IL-13Rα2 = interleukin-13 receptor subunit alpha-2.

## Data Availability

No new data were created or analyzed in this study. Data sharing is not applicable to this article.

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
