# Peer review of "New Era of Immunotherapy in Pediatric Brain Tumors: Chimeric Antigen Receptor T-Cell Therapy"

_ijms, 2021, doi:10.3390/ijms22052404_

Round 1

Reviewer 1 Report

This review article is well-written to provide helpful information in CAR T- cell therapy. Still, there are some recommendations:

In section 2 a general background on the innate and adaptive immunity is recommended to introduce the topic on the five generations of CAR T-cell therapy. A figure detailing the role of the specific immune-mediated molecules involved would be helpful.

In section 3 the immunosuppressive tumor microenvironment side effect is minimally discussed in brain tumors. A more detailed explanation of the effect of the immunosuppressive tumor microenvironment should be argued elucidating the mechanisms by which immune cells organize tumor microenvironments to regulate T-cell activity.

In section 6.1 the adverse effects of CAR-T therapy should be better argued. Please define CRS (Page 7 line 257).

In section 6.2 the authors introduce combinational treatments with PD-1 blockade approaches (Page 8, Line 289). The authors should describe and define the role of immune-checkpoint inhibitors.

In Line 307 the authors should add pieces of information and discuss more in detail the effect on the generation of human memory stem T cells and the mechanism involved.

Author Response

Please see the attachment. Thank you for your comments. 

Reviewer 2 Report

The authors of this manuscript provide a very well-structured overview of the challenges and initial developments of CAR-T cells for the treatment of childhood brain tumors. The abstract summarizes well the main contents of the review. The introduction introduces the field of various relevant childhood brain tumors. Next, the different CAR-T cells are introduced and the functions of the 1st to 5th generation are briefly explained and illustrated. Next, the applications are presented and potential target antigens are introduced. In the following focusses the manuscript on the ongoing preclinical studies. First effects could be shown in suitable mouse models. The main challenge probably remains the identification of further specific target antigens. First clinical studies are rather casuistics and give proof of a basic tolerability. Side effects include cytokine release syndrome and neurotoxicity, both of which must be kept under control. Presumably, the development will lead to combination therapy.

Author Response

(The authors gave the same response as above.)

Round 2

Reviewer 1 Report

The authors have satisfactorily answered point by point to the reviewer’s requests. The data are convincing and clearly presented. Therefore, I suggest the publication of this manuscript.

This manuscript is a resubmission of an earlier submission. The following is a list of the peer review reports and author responses from that submission.